# Association between osteoporosis and periodontal disease among menopausal women: The 2013–2015 Korea National Health and Nutrition Examination Survey

Yunhee Lee [ID] *

Department of Dental Hygiene, Seoyeong University, Paju-si, Gyeonggi-do, Republic of Korea

* yundol79@hanmail.net

## Abstract

### Background

This cross-sectional study aimed to investigate the association between osteoporosis and periodontal disease among Korean menopausal women, as well as the association between osteoporosis and periodontal disease according to duration after menopause.

### Methods

Of a total of 22,948 subjects who participated in the Korea National Health and Nutrition Examination Survey, from 2013 to 2015 the final study population was limited to 2,573 subjects with no missing values. The subjects were divided into two groups, normal bone mineral density (BMD) and osteoporosis, according to the T-score obtained from bone densitometry. Scores of $\geq 3$ points for the community periodontal index of treatment needs were reclassified as periodontal disease. Moreover, after stratification of the variable 'duration after menopause' into 0–4, 5–9, and $\geq 10$ years, binary logistic regression analysis was performed to investigate the association between osteoporosis and periodontal disease according to the duration after menopause.

### Results

There was an association between osteoporosis and periodontal disease. The osteoporosis group had an adjusted odds ratio [OR] of 1.25 (95% confidence interval [CI]: 1.00–1.56) for periodontal disease compared to the normal BMD group. Of note, the osteoporosis group in the menopausal transition stage (0–4 years after menopause) showed an adjusted OR of 2.08 (95% CI: 1.15–3.77) for developing periodontal disease.

### Conclusions

Osteoporosis was associated with periodontal disease and the association was strongest among women in the menopausal transition stage, 0–4 years after menopause. Oral health

**Data Availability Statement:** All relevant data are within the paper and its Supporting information files.

**Funding:** The author received no specific funding for this work.

**Competing interests:** The author have declared that no competing interests exist.

promotion, including regular oral examination and oral hygiene care, is particularly useful for menopausal transition women with osteoporosis.

## Introduction

Osteoporosis is a systemic skeletal disease characterized by the loss of bone mass and micro-architectural deterioration of bone tissue [1]. The level of decrease in bone mineral density (BMD) varies depending on factors such as age and sex; it is typically higher in the elderly population, especially among postmenopausal women [2, 3]. Lifestyle factors associated with low BMD and osteoporosis include alcohol intake, smoking, diabetes, vitamin D insufficiency, low body weight, and physical inactivity [4]. The association between osteoporosis and postmenopausal state is driven by a decrease in estrogen levels forming bone resorbing cytokines, such as RANKL, TNF-α, and interleukin 1, which can contribute to the onset of a series of diseases including osteopenia and osteoporosis [5].

According to the 2020 census by Statistics Korea, the number of Koreans aged 65 years or older is approximately 8.13 million, accounting for 15.7% of the total population. In particular, females account for 76.2% (4.61 million) of the elderly population [6]. Due to an increasingly aging society, osteoporosis has gained increasing social and medical attention.

Menopausal age varies by ethnicity and lifestyle, but the average menopausal age across different countries is commonly reported to be around 50 years [7]. The period between when estrogen secretion starts to decline and up to one year after menopause is referred to as the menopausal transition, which typically starts during the mid-to-late 40s and continues until 4–5 years after the menopause [8]. Sudden systemic BMD loss occurs during the menopausal transition stage, which requires consideration in context of the duration after menopause [9].

Meanwhile, periodontal disease is a chronic inflammatory disease characterized by the destruction of soft and hard tissues that support teeth [10]. Osteoporosis and periodontal disease are multifactorial chronic diseases with systemic and localized bone loss, respectively [11]. According to existing studies, systemic BMD loss due to osteoporosis can also affect alveolar bone density and periodontal disease. In a study on postmenopausal women, the osteoporosis group showed higher clinical attachment loss (CAL) and alveolar bone loss (ABL) than the normal group [12, 13]. In the same context, it has been reported that there is an inverse association between osteoporosis and periodontal disease among postmenopausal women [14, 15]. However, such findings showed different results depending on the progression of periodontal disease and duration after menopause. Accordingly, we investigated the association between osteoporosis and periodontal disease among Korean menopausal women, as well as the association between osteoporosis and periodontal disease according to duration after menopause. The study also aimed to provide basic data for establishing periodontal disease prevention policies for postmenopausal women with osteoporosis.

## Materials and methods

### Study data and study population

This was a cross-sectional study that used raw data from the sixth (2013–2015) Korea National Health and Nutrition Examination Survey (KNHANES-VI) conducted by the Korea Disease Control and Prevention Agency (KDCA). KNHANES is a nationwide survey conducted annually by KDCA that collects national statistics through surveys on health levels, health-related awareness and behavior, and food and nutritional intake. The survey targets approximately

10,000 individuals from a probability sampling of 25 households in 192 regions in Korea [16]. Raw data were obtained from the official KNHANES website (https://knhanes.kdca.go.kr/knhanes/main.do). To limit the study population to menopausal women, natural menopause was defined as 12 consecutive months of amenorrhea without any apparent pathological or physiological cause [17]. Accordingly, the study population was limited to women aged 45–60 years, which is the normal menopausal age range defined by the Korean Society of Menopause, who responded that they had experienced natural menopause. Women who responded that they had artificial menopause due to hysterectomy; individuals who had been previously diagnosed by a physician as having diabetes, which is highly associated with periodontal disease; and individuals with one or more missing responses in the questionnaire were excluded. Among a total of 22,948 participants in KNHANES-VI, 2,573 women were selected as the final study population, who were then divided into a normal BMD group (n = 1,875) and an osteoporosis group (n = 698). This study was conducted with review exemption from the Ministry of Health and Welfare-designated Joint Institutional Review Board (P01-201812-21).

## Variables used in the analysis

For the diagnosis of osteoporosis, a dual-energy radiation absorptiometry was used, and the spine and the head of the femur were examined for body bone density measurement sites. BMD was classified into -1 as normal, -1 to -2.5 as osteopenia, and less than -2.5 as osteoporosis, based on the T-score of the entire femur, neck, and lumbar spine. When calculating the T-score for the prevalence of osteoporosis based on bone densitometry, the prevalence of osteoporosis, which is the variable generated by T-scores calculated using maximum BMD data for Asia (Japan), was used to reclassify the subjects into the normal BMD and osteoporosis groups; this was used as the independent variable. The community periodontal index of treatment needs (CPITN), which was used as the dependent variable, is an index for expressing the need for periodontal treatment that should be provided to all residents or a specific population within a community. CPITN was measured in the examination bus by a public health dentist, and when the results of the simulation training and the field guidance were combined, the degree of agreement was achieved at an appropriate level; the Kappa index of the dental condition was greater than 0.8, and in the periodontal condition was greater than 0.9 [18]. For the presence or absence of periodontal disease, the periodontal tissue condition of the maxillary right molar, maxillary anterior, maxillary left molar, mandibular left molar, mandibular anterior, and mandibular right molar was examined using the periodontal probe. By examining 10 standard teeth of sextant, the highest score in the community periodontal index was recorded, ranging between 0 and 4 points, and reclassified into healthy periodontal status (0–02 points) and periodontal disease (3–4 points) [19]. The study also included general characteristics, such as age (45–50, 51–55, and 56–60 years), education level (elementary school and below, middle school, high school, and college and above), household income level (low, middle-low, middle-upper, and upper), body mass index (normal and obese), and duration after menopause (0–4, 5–9, and $\geq$ 10 years); lifestyle factors, such as alcohol consumption (non-drinker, past drinker, and current drinker) and smoking habit (non-smoker and current smoker); and oral care behavior, such as dental visits within the past year (Yes and No) and frequency of tooth brushing per day ($<$ 3 and $\geq$ 3 times).

## Statistical analysis

KNHANES is a sample of data extracted from the entire Korean population, and was designed using the rolling survey sampling method. KNHANES is a complex sample design survey, and its complex sample design was analyzed after creating a pre-analysis plan file. Kstrata was used

as the stratification variable and enumeration district was used as the clustering variable. For the weight, association analysis weight was used to calculate a new weight to adjust for coverage error according to differences between the sampling frame, households, and population size at the current survey time, and nonresponse error according to non-participation in the survey. During listwise deletion, complex sample information included in the deleted data may be omitted to cause biased estimation of the standard deviation. Accordingly, analysis was performed after generating the interest group and other groups. All analyses were performed using IBM SPSS statistics ver. 22.0 (IBM Corp., Armonk, NY, USA). To identify basic information about the subjects, frequency analysis and descriptive statistics were performed. Moreover, cross-tabulation analysis was performed to identify the presence of osteoporosis according to demographic characteristics, lifestyle factors, and oral care behavior. Furthermore, to investigate the association between osteoporosis and periodontal disease, and the association between osteoporosis and periodontal disease according to duration after menopause, binary logistic regression analysis was performed to calculate an adjusted odds ratio (OR) and 95% confidence interval (CI). Statistical significance was set to p<0.05.

## Results

### BMD groups according to general characteristics, lifestyle factors, and oral care behavior

There were significant differences in osteoporosis status according to age, education level, household income level, BMI, duration after menopause, alcohol consumption, and frequency of tooth brushing per day (p<0.05). With respect to age, the "56–60 years" age group had the highest frequency in both the "normal BMD" and "osteoporosis" groups. With respect to education level, "elementary school or below" was common in the "osteoporosis" group, while all other education levels were common in the "normal BMD" group. With respect to household income level, the "osteoporosis" group had more individuals with a lower education level, while the "normal BMD" group had more individuals with a higher education level. For BMI, there were a high number of individuals with a "normal" BMI in the "normal BMD" group, and a high number of individuals classified as "obese" in the "osteoporosis" group. For duration after menopause, the "osteoporosis" group had a high number of individuals "≥ 10 years" following menopause, while for alcohol consumption, the "normal BMD" group had a high number of "current drinkers." For frequency of tooth brushing per day, the "normal BMD" group had a high number of individuals who brush "≥ 3 times" whereas the "osteoporosis" group had a high number that brush "< 3 times" (Table 1).

### Association between osteoporosis and periodontal disease according to general characteristics, lifestyle factors, and oral care behavior

Table 2 shows the results of the binary logistic regression analysis performed to investigate the association between osteoporosis and periodontal disease. When all other factors were adjusted for, the "osteoporosis group" was associated with an increased risk of periodontal disease compared to the "normal group" (adjusted OR, 1.25; 95% CI: 1.00–1.56). The "51–55 years" age group had an adjusted OR of .79 (95% CI: .64–.99) for developing periodontal disease, as compared to the "56–60 years" age group. The "elementary school or below" and "middle school" groups had an adjusted OR of 1.72 (95% CI: 1.15–2.58) and 1.74 (95% CI: 1.16–2.60) for developing periodontal disease, respectively, as compared to the "college or above" group. The "normal" BMI group had an adjusted OR of .76 (95% CI: .62–.94) for developing periodontal disease, as compared to the "obese" BMI group. Meanwhile, those who responded

Table 1. Bone mineral density (BMD) groups based on general characteristics, lifestyle factors, and oral care behavior (n = 2,573).

| Variables | Category | BMD groups | | | | p-value |
|---|---|---|---|---|---|---|
| | | Normal BMD (n = 1,875) | | Osteoporosis(n = 698) | | |
| Periodontal disease | Healthy | 1,231 | 65.2% | 436 | 63.6% | .512 |
| | Periodontal disease | 644 | 34.8% | 262 | 36.4% | |
| Age | 45–50 | 164 | 11.9% | 14 | 2.9% | .000 |
| | 51–55 | 783 | 43.0% | 226 | 34.4% | |
| | 56–60 | 928 | 45.0% | 458 | 62.7% | |
| Education level | Elementary and below | 608 | 29.0% | 419 | 57.0% | .000 |
| | Middle | 377 | 20.4% | 123 | 18.8% | |
| | High | 603 | 35.1% | 121 | 19.1% | |
| | College and above | 287 | 15.4% | 35 | 5.0% | |
| household income level[a] | Low | 348 | 17.3% | 227 | 29.9% | .000 |
| | Middle low | 502 | 25.2% | 202 | 27.4% | |
| | Middle upper | 464 | 25.3% | 150 | 22.8% | |
| | Upper | 561 | 32.3% | 119 | 19.8% | |
| BMI[b] | Normal | 1,305 | 70.5% | 351 | 50.9% | .000 |
| | Over | 570 | 29.5% | 347 | 49.1% | |
| Duration after menopause(years) | 0–4 | 575 | 35.5% | 85 | 14.1% | .000 |
| | 5–9 | 476 | 26.6% | 131 | 21.8% | |
| | ≥10 | 824 | 37.9% | 482 | 64.1% | |
| Alcohol consumption | No | 423 | 20.8% | 167 | 21.4% | .017 |
| | Past | 380 | 19.8% | 177 | 25.2% | |
| | Dringking | 1,072 | 59.4% | 354 | 53.5% | |
| Smoking habit | No | 1,760 | 93.6% | 655 | 93.3% | .830 |
| | Yes | 115 | 6.4% | 43 | 6.7% | |
| Dental visit <1 year | No | 1,277 | 68.1% | 495 | 70.6% | .295 |
| | Yes | 598 | 31.9% | 203 | 29.4% | |
| Frequency of tooth brushing per day | <3 times | 921 | 46.9% | 402 | 56.5% | .000 |
| | ≥3 times | 954 | 53.1% | 296 | 43.5% | |

Values are presented as n (weighted %).

*p<0.05,

**p<0.01,

***p<0.001

[a]Income quartile.

[b]Asia-Pacific Standard: normal,<25kg/m$^2$; obese,≥25 kg/m$^2$.

"non-smoker" for smoking status and "No" to dental visits within the past year had an adjusted OR of .63 (95% CI: .42–.94) and .94 (95% CI: .76–1.16) for developing periodontal disease, respectively.

## Association between osteoporosis and periodontal disease according to duration after menopause

Table 3 shows the results of the binary logistic regression analysis performed after stratification of the variable "duration after menopause (years)" to investigate the association between osteoporosis and periodontal disease according to the duration after menopause. When all other factors were adjusted for, a duration after menopause (years) of "0–4 years" in the

**Table 2. Association between osteoporosis and periodontal disease according to general characteristics, lifestyle factors, and oral care behavior (n = 2,573).**

| Variables | Category | Periodontal disease | |
|---|---|---|---|
| | | aOR | (95% CI) |
| Osteoporosis | Normal | 1.00 | |
| | Osteoporosis | 1.25 | (1.00–1.56)* |
| Age | 45–50 | .68 | (0.43–1.07) |
| | 51–55 | .79 | (0.64–0.99)* |
| | 56–60 | 1.00 | |
| Duration after menopause(years) | 0–4 | 0.97 | (0.74 1.27) |
| | 5–9 | 1.00 | (0.78–1.29) |
| | ≥10 | 1.00 | |
| Education level | Elementary and below | 1.72 | (1.15–2.58)** |
| | Middle | 1.74 | (1.16–2.59)** |
| | High | 1.01 | (0.70–1.46) |
| | College and above | 1.00 | |
| household income level[a] | Low | 1.31 | (0.96–1.78) |
| | Middle low | 1.10 | (0.83–1.45) |
| | Middle upper | 0.96 | (0.72–1.29) |
| | Upper | 1.00 | |
| BMI[b] | Normal | 0.76 | (0.62–0.94)** |
| | Over | 1.00 | |
| Alcohol consumption | No | 1.14 | (0.89 1.45) |
| | Past | 0.95 | (0.75–1.20) |
| | Drinking | 1.00 | |
| Smoking habit | No | 0.63 | (0.42–0.94)* |
| | Yes | 1.00 | |
| Dental visit <1 year | No | 0.94 | (0.76–1.16) |
| | Yes | 1.00 | |
| Frequency of tooth brushing per day | <3 times | 1.09 | (0.89–1.35) |
| | ≥3 times | 1.00 | |

Adjusted for age, education level, household income level, BMI, duration after menopause(years), alcohol consumption, smoking habit, dental visit <1 years, frequency of tooth brushing per day.

*p<0.05,

**p<0.01,

aOR: adjusted odds ratio, CI: confidence interval.

[a]Income quartile.

[b]Asia-Pacific Standard: normal,<25kg/m$^2$; obese,≥25 kg/m$^2$.

**Table 3. Association between osteoporosis and periodontal disease according to duration after menopause (years).**

| Variables | Stratum | Healthy | Periodontal disease | |
|---|---|---|---|---|
| | | aOR | aOR | (95% CI) |
| Duration after menopause(years) | 0-4a | 1.00 | 2.08 | (1.15–3.77)* |
| | 5-9b | 1.00 | 1.62 | (0.97–2.71) |
| | ≥10c | 1.00 | 1.00 | (0.76–1.32) |

Adjusted for age, education level, household income level, BMI, alcohol consumption, smoking habit, dental visit <1 years, frequency of tooth brushing per day.

*p<0.05, aOR: adjusted odds ratio, CI: confidence interval.

"osteoporosis group" was associated with an increased risk of periodontal disease compared to the equivalent duration after menopause in the "normal BMD group" (adjusted OR, 2.08; 95% CI: 1.15–3.77).

## Discussion

In this study, which analyzed natural postmenopausal women aged 45–60 years correcting for various potential factors related to periodontal disease, a correlation was identified between osteoporosis and periodontal disease as well as between osteoporosis and periodontal disease according to the postmenopausal period. Individuals with osteoporosis were found to be more likely to develop periodontal disease, supporting the results of previous studies. Mashalkar et al. [13] used dual-energy X-ray absorptiometry (DXA) to measure the BMD of 98 Indian menopausal women aged 45–65 years and found that periodontal pocket depth (PD) and CAL were higher in the osteoporosis group. In a study by Penoni et al. [14] including 134 Brazilian menopausal women aged 65–80 years, the osteoporosis group showed a 2.49 higher risk of periodontal disease and higher gingival recession and CAL than the normal group. In a 6-year prospective study by Mau et al. [12] using Taiwanese national health insurance data, the risk of mild, severe, and mild periodontitis was 1.56, 2.09, and 2.08 times higher in the osteoporosis group, respectively. In a similar context, it can be viewed that osteoporosis may worsen periodontal status. However, a patient-control study by Ayed et al. [20] of Saudi Arabian postmenopausal women aged 50–70 years assessed periapical dental radiographs and reported that there were differences in PD and CAL between the osteoporosis and normal groups, but only the difference in CAL was statistically significant. A study by Sultan et al. [21], which assessed the association between periodontal disease and systemic BMD of Indian postmenopausal women aged ≥ 50 years, reported that osteoporosis is not associated with average ABL and CAL. Such conflicting study results could be attributed to differences in the progression of periodontal disease, age of subjects, and duration after menopause, as well as different clinical tools used to assess periodontal status and osteoporosis.

Meanwhile, it has been reported that osteopenia, osteoporosis, and other related risks are higher during the menopausal transition due to the presence of a sudden estrogen deficiency. A 20-year prospective study by Karlamangla et al. [22] involving 2,000 American women reported that the measurement of BMD using DXA showed that bone loss increased sharply between one year prior to and two years after menopause, regardless of race and ethnicity, but bone loss decreased thereafter. In a prospective study by Sowers et al. that investigated 918 American, Chinese, and Japanese women over an 8-year period, bone loss increased sharply starting from two years prior to menopause and reached the peak level by approximately 1–1.5 years after the onset of menopause, but by approximately 4–6 years after menopause it was maintained at a relatively stable level [23]. In a 15-year prospectively study by Sowers et al. on 629 American women, measurement of BMD using DXA showed that bone loss was 1.7% per year during three years prior to menopause, which decreased to 3.3% per year during two years after menopause and then to 1.1% per year for the following five years [24]. In the present study, the risk of the periodontal disease appeared more prominently in menopausal transition women with osteoporosis who were 0–4 years of duration after menopause, which supported the findings of previous studies. Such results could be interpreted as the risk of developing periodontal disease being more prominent during the menopausal transition stage, due to a decrease in BMD caused by sudden estrogen deficiency.

In this study, the OR of developing the periodontal disease was more prominent in those with lower education levels, while those in the osteoporosis group who were younger, had a

normal weight, were non-drinkers, and non-smokers showed a lower OR for developing periodontal disease. Such results are related to the study by Rezaei et al. [25] reporting that people belonging to groups with lower education levels had lower dental care utilization rates. Such results appeared consistently despite different methods used to measure socioeconomic status and having different study periods and subjects, which is assumed to be the result of people with high education levels having a higher rate of early treatment owing to easier access to healthcare. In 1996, the International Academy of Periodontology designated *A. actinomycetemcomitans*, *P. gingivalis*, and *T. forsythensis* as pathogens that cause destructive periodontal disease [26]. In a 15-year prospective study by Feres et al. [27], pathogens that cause periodontal disease, such as *P. gingivalis*, were detected at the highest frequency among the elderly; such results indicate that periodontal disease increases with increasing age. Moreover, the self-cleaning action of saliva also decreases and oral hygiene ability declines with an increase in age, which could accelerate periodontal disease due to the faster accumulation of biofilm [28]. Periodontal disease is an infectious disease, but personal habits, such as drinking and smoking, and systemic factors, such as body weight, also have an influence. Drinking and smoking have been considered causes for periodontal disease according to some authors, and carry a high risk of periodontal destruction and tooth loss. Drinking has a negative effect on maintaining the balance of healthy oral microflora and alters the pH of saliva to cause an increase in pathogenic bacteria with no acid resistance, such as *Aggregatibacter actincetemcomitans* [29, 30]. Smoking is known to exacerbate periodontal disease by lowering gingival blood flow and promoting constriction of peripheral blood vessels by increasing the release of epinephrine [31]. Previous studies have also reported an association between BMI and periodontal disease. A study by Genco et al. used data from the US National Health and Nutrition Examination Survey III (NHANES-III) to investigate 12,367 non-diabetic subjects aged up to 90 years and found that being obese was associated with severe periodontitis, showing a significant association with OR value of 1.45 (1.09–1.93) after adjusting for all confounding variables [32].

This work represents an attempt to conduct a representative study using data from KNHANES, which objectively measured periodontal status according to the WHO criteria. Despite such efforts, the study had the limitation of not being able to identify the causal relationship between the influencing factors and periodontal disease, due to it being a cross-sectional study. In addition, periodontal disease was evaluated only with the CPI without using other periodontal-related indicators. The significance of this study can be found in the fact that it adjusted for various variables, including age, duration after menopause, lifestyle factors, and oral care behavior; that it used CPI, which is an objective scale for measuring periodontal tissue status; and that it analyzed the association between osteoporosis and periodontal disease stratified by duration after menopause and by considering a more diverse set of factors.

## Conclusion

The objective of this study was to investigate the association between osteoporosis and periodontal disease among Korean menopausal women based on raw data from KNHANES-VI (2013–2015) conducted by KDCA. CPI, which is an objective scale for measuring periodontal tissue status, was used to more clearly identify the association between osteoporosis and periodontal disease by adjusting for general characteristics, lifestyle factors, and oral care behavior. The results showed an association between osteoporosis and the risk of periodontal disease. Moreover, such effects appeared more prominently among menopausal transition women 0–4 years after menopause. Oral health promotion, including regular oral examination and oral hygiene care, is therefore particularly useful for menopausal transition women with osteoporosis in the context of establishing health policies for postmenopausal women with osteoporosis.

## Supporting information

**S1 File.**
(ZIP)

## Author Contributions

**Conceptualization:** Yunhee Lee.

**Data curation:** Yunhee Lee.

**Formal analysis:** Yunhee Lee.

**Funding acquisition:** Yunhee Lee.

**Investigation:** Yunhee Lee.

**Methodology:** Yunhee Lee.

**Project administration:** Yunhee Lee.

**Resources:** Yunhee Lee.

**Software:** Yunhee Lee.

**Supervision:** Yunhee Lee.

**Validation:** Yunhee Lee.

**Visualization:** Yunhee Lee.

**Writing – original draft:** Yunhee Lee.

**Writing – review & editing:** Yunhee Lee.

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
