## [Decision Letter · Decision Letter 0]

26 Oct 2021

PONE-D-21-25215Association between Osteoporosis and Periodontal disease among Menopausal transition women : The 2013-2015 Korea National Health and Nutrition Examination SurveyPLOS ONE

Dear Dr. lee,

Thank you for submitting your manuscript to PLOS ONE. After careful consideration, we feel that it has merit but does not fully meet PLOS ONE’s publication criteria as it currently stands. Therefore, we invite you to submit a revised version of the manuscript that addresses the points raised during the review process.

The manuscript needs to be revised according to reviewers comments and suggestions.

We look forward to receiving your revised manuscript.

Kind regards,

Kunaal Dhingra, MDS, MFDS RCPS (Glasg), MFDS RCS (Eng), MDTFEd

Academic Editor

PLOS ONE

Journal Requirements: 

3. Please ensure that you include a title page within your main document. You should list all authors and all affiliations as per our author instructions and clearly indicate the corresponding author

Reviewers' comments:

Reviewer #1: I hope that the comments below could help the authors to improve and adjust their work, whether for this or another journal.

English Language: In general, the text should be revised with the intention of making the reading more fluid. Although the manuscript is understandable, it would benefit a lot from careful editing by an expert in writing scientific English.

Title: Well written, makes the reader interested in the full article.

Keywords: I did not find the term “Menopausal transition” in Mesh. I suggest “Menopause”, instead. I also suggest inserting the term “Korea National Health and Nutrition Examination Survey” among the keywords.

Abstract

1. Methods: I suggest inserting the study design: “this cross-sectional study aimed...”This sentence, “formerly Korea Centers for Disease Control and Prevention”, is not necessary for the Abstract; just in the Method section. I suggest substitute it for the period of the KNHANES, eg: “...conducted by the Korea Disease Control and Prevention Agency (KDCA) from 2013 to 2015”.

Please, simplify this sentence, clarifying: “When calculating the T-score for the prevalence of osteoporosis based on bone densitometry, the prevalence of osteoporosis, which is the variable generated by T-scores calculated using maximum bone mineral density data for Asia (Japan), was used to divide the subjects into the normal and osteoporosis groups.” Suggestion: The subjects were divided into two groups, normal and osteoporosis, according to the T-score obtained from bone densitometry.

It’s important to add information on the variable ‘duration after menopause’, categorized as 0-4, 5-9, and ≥ 10 years.

Please, also add information on statistic analysis.

2- Results: The authors state that “The osteoporosis group had an adjusted OR of 1.25 (95% CI:1.00-1.56) for developing periodontal disease”. The term “for developing” could be used if the study had a longitudinal design. With a cross-sectional design, authors should limit their statement to “for presenting periodontal disease”. The same for the next line of Results.

Conclusions: The term “risk factor” in Epidemiology is frequently involved with causality and is better linked to cohort studies. I suggest that “risk factor”, in “Osteoporosis could be considered a risk factor of periodontal disease”, should be substituted by “Osteoporosis was an associated factor with periodontal disease”, or “Osteoporosis could be considered a risk indicator of periodontal disease in the studied population. Such effect was...”

Introduction

I recommend the authors restructure/re-elaborate their introduction.

Unfortunately, the English language shows low quality. Again, I strongly recommend careful proofreading. Throughout the whole manuscript, the misuse, misspellings are noted.

The first paragraph should introduce the main topic involved in the study: osteoporosis. So, the second paragraph could be the first one, followed by the others information.

In the 2nd paragraph, “The level of decrease in bone mineral density (BMD) may vary depending on age and gender, but it is typically accelerated among the older population than the younger population and among women, especially postmenopausal women, than men”, could be simplified by “The level of decrease in bone mineral density (BMD) vary depending on factors such as age and gender *, and is typically higher in the elderly population, especially among postmenopausal women.

* Not only age and gender are related to the decrease in BMD. Factors like smoking, diabetes, low body mass index, glucocorticoids use, rheumatoid arthritis, and others, are related to low BMD

Following the sentence above: “The cause of this is related to a postmenopausal decrease in estrogen level forming bone resorbing cytokines, such as RANKL, TNF-α, and interleukin 1, which could contribute to the onset of a series of diseases, including osteopenia and osteoporosis”. Then, replace the first paragraph on following: " According to the 2020 census by Statistics Korea, the population size of Koreans aged 65 years or older is approximately 8.13 million, accounting for 15.7% of the total population. In particular, females account for 76.2% of the elderly population with 4.61 million. As a result of reaching such aging society, osteoporosis has garnered social and medical attention.”

Line 47: “Osteoporosis and periodontal disease are multifactorial chronic diseases with

systemic and localized bone loss”, add the term “respectively”, since there is no evidence that periodontal disease causes systemic bone loss.

Lines 53 and 54: “this study used raw data from the sixth (2013-2015) Korea National Health and Nutrition Examination Survey (KNHANES-VI) conducted by the Korea Disease Control and Prevention Agency (KDCA)”: this belongs to Materials and Methods.

Materials and Methods:

1-Please, begin this section with the study design. “This was a cross-sectional study...”

2-Variables used in the analysis:

About the periodontal exam: Please, describe how many examiners were involved in the data collection. Were they calibrated? Is there a measure of reliability, such as the kappa coefficient? Where the exams were performed (at dental offices?)

Please, provide more details on CPITN: all present teeth were examined and the higher observed CPITN score was attributed to the sextant, or only the index teeth were examined and the higher observed CPITN score attributed to the sextant?

-The assessment of bone mass needs to be clarified. Was the densitometry performed using Dual-energy X-ray absorptiometry (DXA)? The authors do not define the T-score used to classify osteoporosis. Was it ≤-2,5? Which bone sites were considered: lumbar spine, femoral neck, and total fêmur?

3-Statistical analysis: Please, clarify lines 94 to 97.

Information on the statistical tests used is lacking. Example, in table 2: Kruskal-Wallis, Mann Whitney, Qui-Square?

Results:

- General characteristics of subjects: please, focus on the main results. It is not recommended to repeat every data already shown in table 1.

- Tables 1 and 2: women were divided into 2 groups, according to BMD: normal BMD and osteoporosis. Please, substitute the name of the variable in the upper cells “osteoporosis” for “BMD groups”. Then, separating normal BMD x osteoporosis will be ok.

- I would appreciate seeing the percentage of each score of CPITN in this table.

- Title of table 2. Osteoporosis Based on General Characteristics(n=2,048).

Again, it was not osteoporosis. It was the bone mineral density and the according to groups of normal BMD x osteoporosis.

I did not understand the 2,048 if the sum is 2,573.

- Association between osteoporosis and periodontal disease

Please, the authors should express, in the text, what is the meaning of the OR below 1.0, such as: .79 (95% CI:.64-.99), .76 (95% CI:.62-.94), .63 (95% CI:.42-.94) and .94 (95% CI:.76-1.16) for developing periodontal disease. Would these factors be protective factors? Which percentage?

- Tables 3 and 4: please, the titles could be more elaborated.

Discussion:

Line 157: Please, I recommend start the Discussion in a different manner, eg: Based on the analysis of naturally menopausal women aged 45-60 years in this study, ....

Line 163: [14] was not a prospective study.

Line 165 [12]; please, the investigation of this report was performed in another way. Check the results and conclusion and re-elaborate the sentence.

Line 185: decreased or increased?

Line 200 and others: the authors need to write the name of the authors included in the manuscript in a standard way.

The higher the age, the higher the systemic bone loss. Why would the duration 0-4 years be associated with periodontal disease, and the higher periods not? This is the main topic that should be involved in this section. Please, discuss this important finding.

Line 205: “Drinking and smoking are key causes of periodontal disease” Please, the authors should not state that drinking is a key factor because it is not evidenced as a risk factor for periodontal disease. I recommend they should be more careful and, at the maximum, state that drinking may be an associated factor related to periodontal disease. Another way is “Drinking and smoking were considered causes for periodontal disease according to some authors...” Drinking has shown (instead of ‘has’) a negative effect on maintaining the balance...”

Line 209: Aggregatibacter. Please, correct.

Limitations:

1- The cut-off for osteoporosis was based on a T-score (the authors should provide information on this in Materials and Methods). This means that the intermediate scores, even those representing advanced osteopenia, close to osteoporosis, were included in the normal BMD group. Considering osteopenia, even severe cases, as a normal group, is a limitation of the study.

Explore limitations of the CPITN.

I missed the suggestion of new studies.

Reviewer #2: This analysis uses the KNHANES surveillance dataset to estimate the association of osteoporosis and periodontitis among natural post-menopausal women aged 45-60. The analysis is well done in general, but I would suggest a few changes to improve the manuscript. First, the abstract should mention that the models controlled for confounding variables. Next, I encourage the author to run an unweighted analysis and compare it to the weighted analysis. If they are very close, I would recommend just presenting the unweighted analysis. It is unclear if the efforts to weight an analysis of such a small subgroup is a good idea or not. Third, it would be easier to interpret if Tables 1 and 2 were combined (author could include an “all” column in Table 2 to combine both tables). Fourth, the covariates selected seem appropriate, but the models are not fitted, in that the final models contain variables that do not contribute to model fit. I recommend using stepwise selection to fit the model and ensure optimal model fit (see Bursac Z, Gauss CH, Williams DK, Hosmer DW. Purposeful selection of variables in logistic regression. Source code for biology and medicine. 2008 Dec;3(1):1-8.). Fifth, the style of presentation of the model (where just the exposure variables of interest have their ORs reported) is ideal in Table 4, and my suggestion would be to combine tables 3 and 4 and only present the ORs for the exposure variables (and not interpret them – just for the sake of brevity). The author can add a note indicating which confounders are also in the model that survive the stepwise modeling process. The author should make sure to call out the tables in the text when mentioning numbers. The section “Osteoporosis according to general characteristics…” should contain some estimates to support the claims (like the other sections do).

The p-values reported in tables appear to be chi-square, but this is not made clear, and if Fisher’s exact test is used, this is also not clear. When interpreting the chi-square results, it is important to mention direction of association. The discussion is very good in that it interprets the results against the existing literature, but there are some awkward sentences. I will recommend minor corrections and then when I see the next version, I will give more careful feedback about the discussion.

---

## [Author Response · Author response to Decision Letter 0]

7 Dec 2021

Reviewer #1

Page Line Comment Response

 English Language: In general, the text should be revised with the intention of making the reading more fluid. Although the manuscript is understandable, it would benefit a lot from careful editing by an expert in writing scientific English. Thank you for highlighting this issue. I agree with this and the manuscript has now been edited by a native English speaker who is an expert in writing scientific English.

1 1 Title: Well written, makes the reader interested in the full article.

Keywords: I did not find the term “Menopausal transition” in Mesh. I suggest “Menopause”, instead. I also suggest inserting the term “Korea National Health and Nutrition Examination Survey” among the keywords. Agree. I have made the suggested changes.

1

4 21-23

77-79

 Methods: I suggest inserting the study design: “this cross-sectional study aimed...”This sentence, “formerly Korea Centers for Disease Control and Prevention”, is not necessary for the Abstract; just in the Method section. I suggest substitute it for the period of the KNHANES, eg: “...conducted by the Korea Disease Control and Prevention Agency (KDCA) from 2013 to 2015”. Thank you. I have made the changes you have suggested.

2 40-42 It’s important to add information on the variable ‘duration after menopause’, categorized as 0-4, 5-9, and ≥ 10 years.

Please, also add information on statistic analysis. I agree with you and have made the required changes.

2 39-42 2- Results: The authors state that “The osteoporosis group had an adjusted OR of 1.25 (95% CI:1.00-1.56) for developing periodontal disease”. The term “for developing” could be used if the study had a longitudinal design. With a cross-sectional design, authors should limit their statement to “for presenting periodontal disease”. The same for the next line of Results. I have made the required changes.

2 39-42 Conclusions: The term “risk factor” in Epidemiology is frequently involved with causality and is better linked to cohort studies. I suggest that “risk factor”, in “Osteoporosis could be considered a risk factor of periodontal disease”, should be substituted by “Osteoporosis was an associated factor with periodontal disease”, or “Osteoporosis could be considered a risk indicator of periodontal disease in the studied population. Such effect was...” I have changed the text in light of your comments.

2

3 43-

74 Introduction

I recommend the authors restructure/re-elaborate their introduction.

Unfortunately, the English language shows low quality. Again, I strongly recommend careful proofreading. Throughout the whole manuscript, the misuse, misspellings are noted. Thank you. The Introduction has been rewritten and the whole manuscript has been edited by a professional English language Editor.

2 44 The first paragraph should introduce the main topic involved in the study: osteoporosis. So, the second paragraph could be the first one, followed by the others information. I agree with you. The Introduction has been rewritten as per your suggestion.

2 44- * Not only age and gender are related to the decrease in BMD. Factors like smoking, diabetes, low body mass index, glucocorticoids use, rheumatoid arthritis, and others, are related to low BMD

Following the sentence above: “The cause of this is related to a postmenopausal decrease in estrogen level forming bone resorbing cytokines, such as RANKL, TNF-α, and interleukin 1, which could contribute to the onset of a series of diseases, including osteopenia and osteoporosis”. Then, replace the first paragraph on following: " According to the 2020 census by Statistics Korea, the population size of Koreans aged 65 years or older is approximately 8.13 million, accounting for 15.7% of the total population. In particular, females account for 76.2% of the elderly population with 4.61 million. As a result of reaching such aging society, osteoporosis has garnered social and medical attention.” I have edited the manuscript to address your comments.

3 63 “Osteoporosis and periodontal disease are multifactorial chronic diseases with

systemic and localized bone loss”, add the term “respectively”, since there is no evidence that periodontal disease causes systemic bone loss. Thank you for highlighting this issue, which has been addressed in the revised manuscript.

4 77 “this study used raw data from the sixth (2013-2015) Korea National Health and Nutrition Examination Survey (KNHANES-VI) conducted by the Korea Disease Control and Prevention Agency (KDCA)”: this belongs to Materials and Methods. I agree and have moved the text to the Materials and Methods section accordingly.

4 77 1-Please, begin this section with the study design. “This was a cross-sectional study...” This change has now been made.

5 105 About the periodontal exam: Please, describe how many examiners were involved in the data collection. Were they calibrated? Is there a measure of reliability, such as the kappa coefficient? Where the exams were performed (at dental offices?)

Please, provide more details on CPITN: all present teeth were examined and the higher observed CPITN score was attributed to the sextant, or only the index teeth were examined and the higher observed CPITN score attributed to the sextant? This detail has been elaborated on in the revised manuscript.

4 96 The assessment of bone mass needs to be clarified. Was the densitometry performed using Dual-energy X-ray absorptiometry (DXA)? The authors do not define the T-score used to classify osteoporosis. Was it ≤-2,5? Which bone sites were considered: lumbar spine, femoral neck, and total fêmur? Thank you for your comments. The manuscript has been revised to address these questions.

5 121 Statistical analysis: Please, clarify lines 94 to 97.

Information on the statistical tests used is lacking. Example, in table 2: Kruskal-Wallis, Mann Whitney, Qui-Square? The statistical tests are now elucidated on in the revised manuscript.

9 170 Association between osteoporosis and periodontal disease

Please, the authors should express, in the text, what is the meaning of the OR below 1.0, such as: .79 (95% CI:.64-.99), .76 (95% CI:.62-.94), .63 (95% CI:.42-.94) and .94 (95% CI:.76-1.16) for developing periodontal disease. Would these factors be protective factors? Which percentage? The text has been revised to clarify the results.

6

8

10 150

166

186 please, the titles could be more elaborated. The titles have been changed accordingly.

11 199 Discussion:

Line 157: Please, I recommend start the Discussion in a different manner, eg: Based on the analysis of naturally menopausal women aged 45-60 years in this study, ....

Line 163: [14] was not a prospective study.

Line 165 [12]; please, the investigation of this report was performed in another way. Check the results and conclusion and re-elaborate the sentence.

Line 185: decreased or increased?

Line 200 and others: the authors need to write the name of the authors included in the manuscript in a standard way. Thank you for this comment. I have revised the Discussion section accordingly.

12 218-232 The higher the age, the higher the systemic bone loss. Why would the duration 0-4 years be associated with periodontal disease, and the higher periods not? This is the main topic that should be involved in this section. Please, discuss this important finding. Thank you for this insightful comment. To answer your question, in individuals who are 3–4 years after menopause, it is hypothesized that the increased risk of periodontal disease may be due to a decrease in bone density due to a sudden lack of estrogen. This has been mentioned as a possible causative factor in the manuscript.

13 247 Line 205: “Drinking and smoking are key causes of periodontal disease” Please, the authors should not state that drinking is a key factor because it is not evidenced as a risk factor for periodontal disease. I recommend they should be more careful and, at the maximum, state that drinking may be an associated factor related to periodontal disease. Another way is “Drinking and smoking were considered causes for periodontal disease according to some authors...” Drinking has shown (instead of ‘has’) a negative effect on maintaining the balance...” I agree with you and have modified the manuscript text accordingly.

13 250 Line 209: Aggregatibacter. Please, correct. This has been modified as requested.

13 258 Limitations:

1- The cut-off for osteoporosis was based on a T-score (the authors should provide information on this in Materials and Methods). This means that the intermediate scores, even those representing advanced osteopenia, close to osteoporosis, were included in the normal BMD group. Considering osteopenia, even severe cases, as a normal group, is a limitation of the study.

Explore limitations of the CPITN. The limitations of the study have been expanded on in the revised manuscript, as per your suggestion.

1 15 the abstract should mention that the models controlled for confounding variables. This has been addressed in the revised abstract.

 I encourage the author to run an unweighted analysis and compare it to the weighted analysis. If they are very close, I would recommend just presenting the unweighted analysis. It is unclear if the efforts to weight an analysis of such a small subgroup is a good idea or not. Thank you for pointing this out. I agree with this comment. Comparing the unweighted analysis to the weighted equation, the results were very similar.

7 165 it would be easier to interpret if Tables 1 and 2 were combined (author could include an “all” column in Table 2 to combine both tables). Table 1 and 2 seemed to overlap, so Table 1 was deleted and Table 2 was modified.

10 194 the style of presentation of the model (where just the exposure variables of interest have their ORs reported) is ideal in Table 4, and my suggestion would be to combine tables 3 and 4 and only present the ORs for the exposure variables (and not interpret them – just for the sake of brevity). In Table 4, a binary logistic regression analysis was performed after stratifying the ‘postmenopausal period (years)’ variable to understand the relationship between osteoporosis and periodontal disease according to the postmenopausal period, separate from Table 3.

11 198 The discussion is very good in that it interprets the results against the existing literature, but there are some awkward sentences. I will recommend minor corrections and then when I see the next version, I will give more careful feedback about the discussion. Thank you for your comment. I have revised the discussion, which has been edited by a professional English language Editor. I hope that the revised discussion is acceptable.

---

## [Decision Letter · Decision Letter 1]

8 Feb 2022

PONE-D-21-25215R1Association between osteoporosis and periodontal disease among menopausal women:

The 2013-2015 Korea National Health and Nutrition Examination SurveyPLOS ONE

Dear Dr. lee,

Thank you for submitting your manuscript to PLOS ONE. After careful consideration, we feel that it has merit but does not fully meet PLOS ONE’s publication criteria as it currently stands. Therefore, we invite you to submit a revised version of the manuscript that addresses the points raised during the review process.

Kindly modify your manuscript according to reviewer suggestions.

We look forward to receiving your revised manuscript.

Kind regards,

Kunaal Dhingra, MDS, MFDS RCPS (Glasg), MFDS RCS (Eng), MDTFEd

Academic Editor

PLOS ONE

Journal Requirements:

Reviewers' comments:

Reviewer #1: 

The authors made many efforts to achieve what is expected for a paper published in this prestigious journal They have followed many of the recommendations and restructured/re-elaborated their manuscript in a satisfactory way.

I would just recommend minor adjustments, as shown below:

Abstract

1. Methods: I suggest inserting the period of the KNHANES, “Korea National Health and Nutrition Examination Survey, from 2013 to 2015”, the final population...

Conclusions: I suggest using past tense instead of the present tense: “Osteoporosis was an associated factor with periodontal disease, and the association...”

Introduction

Line 57:“According to existing studies, systemic BMD loss due to osteoporosis can also affect alveolar bone density and periodontal disease. In a study on postmenopausal women...” Line 60 “In the same context, it has been reported that there is an association between osteoporosis and periodontal disease among postmenopausal women ...”

These sentences bring the same information. It seems to me that the authors wanted to talk about the bidirectional via between osteoporosis and periodontal disease. If so, in line 60/61, it has to become clear, maybe inserting the term “inverse”, eg “it has been reported that there is an inverse association between osteoporosis and periodontal disease...”

Results: Please, correct terms in footnotes, like “alchol” and “consumpion”. Besides, some parenthesis is missing in the table

Discussion:

Line 239: Please, correct “CPI”

Reviewer #2: This paper was already very good, in that the methods were sound and statistical analysis was appropriate. The revisions made by the authors improved it.

---

## [Author Response · Author response to Decision Letter 1]

22 Feb 2022

Methods: I suggest inserting the period of the KNHANES, “Korea National Health and Nutrition Examination Survey, from 2013 to 2015”, the final population... I agree with you and have made the required changes.

Conclusions: I suggest using past tense instead of the present tense: “Osteoporosis was an associated factor with periodontal disease, and the association...” I agree with you and have made the required changes.

Line 57:“According to existing studies, systemic BMD loss due to osteoporosis can also affect alveolar bone density and periodontal disease. In a study on postmenopausal women...” Line 60 “In the same context, it has been reported that there is an association between osteoporosis and periodontal disease among postmenopausal women ...”

These sentences bring the same information. It seems to me that the authors wanted to talk about the bidirectional via between osteoporosis and periodontal disease. If so, in line 60/61, it has to become clear, maybe inserting the term “inverse”, eg “it has been reported that there is an inverse association between osteoporosis and periodontal disease...” I agree with you and have made the required changes.

Results: Please, correct terms in footnotes, like “alchol” and “consumpion”. Besides, some parenthesis is missing in the table I agree with you and have made the required changes.

Discussion:

Line 239: Please, correct “CPI” Thank you. I have made the changes you have suggested.

---

## [Decision Letter · Decision Letter 2]

7 Mar 2022

Association between osteoporosis and periodontal disease among menopausal women:

The 2013-2015 Korea National Health and Nutrition Examination Survey

PONE-D-21-25215R2

Dear Dr. lee,

We’re pleased to inform you that your manuscript has been judged scientifically suitable for publication and will be formally accepted for publication once it meets all outstanding technical requirements.

Kind regards,

Kunaal Dhingra, MDS, MFDS RCPS (Glasg), MFDS RCS (Eng), MDTFEd

Academic Editor

PLOS ONE

---

## [Editor Report · Acceptance letter]

10 Mar 2022

PONE-D-21-25215R2 

Association between osteoporosis and periodontal disease among menopausal women:
The 2013-2015 Korea National Health and Nutrition Examination Survey 

Dear Dr. Lee:

I'm pleased to inform you that your manuscript has been deemed suitable for publication in PLOS ONE. Congratulations! Your manuscript is now with our production department. 

Kind regards, 

on behalf of

Dr. Kunaal Dhingra 

Academic Editor

PLOS ONE